# Robust Anytime Learning of Markov Decision Processes [*]

**Marnix Suilen**
Department of Software Science
Radboud University
Nijmegen, The Netherlands

**Thiago D. Simão**
Department of Software Science
Radboud University
Nijmegen, The Netherlands

**David Parker**
Department of Computer Science
University of Oxford
Oxford, United Kingdom

**Nils Jansen**
Department of Software Science
Radboud University
Nijmegen, The Netherlands

## Abstract

Markov decision processes (MDPs) are formal models commonly used in sequential decision-making. MDPs capture the stochasticity that may arise, for instance, from imprecise actuators via probabilities in the transition function. However, in data-driven applications, deriving precise probabilities from (limited) data introduces statistical errors that may lead to unexpected or undesirable outcomes. Uncertain MDPs (uMDPs) do not require precise probabilities but instead use so-called uncertainty sets in the transitions, accounting for such limited data. Tools from the formal verification community efficiently compute robust policies that provably adhere to formal specifications, like safety constraints, under the worst-case instance in the uncertainty set. We continuously learn the transition probabilities of an MDP in a robust anytime-learning approach that combines a dedicated Bayesian inference scheme with the computation of robust policies. In particular, our method (1) approximates probabilities as intervals, (2) adapts to new data that may be inconsistent with an intermediate model, and (3) may be stopped at any time to compute a robust policy on the uMDP that faithfully captures the data so far. Furthermore, our method is capable of adapting to changes in the environment. We show the effectiveness of our approach and compare it to robust policies computed on uMDPs learned by the UCRL2 reinforcement learning algorithm in an experimental evaluation on several benchmarks.

## 1 Introduction

Sequential decision-making in realistic scenarios is inherently subject to uncertainty, commonly captured via probabilities. *Markov decision processes* (MDPs) are the standard model to reason about such decision-making problems [Puterman, 1994, Bertsekas, 2005]. Safety-critical scenarios require assessments of correctness which can, for instance, be described by temporal logic [Pnueli, 1977] or expected reward specifications. A fundamental requirement for providing such correctness guarantees on MDPs is that probabilities are precisely given. Methods such as variants of model-based reinforcement learning [Moerland et al., 2020] or PAC-learning [Strehl et al., 2009] can learn MDPs by deriving *point estimates* of probabilities from data to satisfy this requirement. This

---

[*]Supported by NWO OCENW.KLEIN.187: "Provably Correct Policies for Uncertain Partially Observable Markov Decision Processes", NWA.1160.18.238 (PrimaVera) and by the ERC under the European Union's Horizon 2020 research and innovation programme (FUN2MODEL, grant agreement No. 834115).

36th Conference on Neural Information Processing Systems (NeurIPS 2022).

derivation naturally carries the risk of statistical errors. Optimal policies are highly sensitive to small perturbations in transition probabilities, leading to sub-optimal outcomes such as a deterioration in performance [Mannor et al., 2007, Goyal and Grand-Clement, 2020].

*Uncertain MDPs* (uMDPs; also known as *interval* or *robust* MDPs) extend MDPs to incorporate such statistical errors by introducing an additional layer of uncertainty via *uncertainty sets* on the transition function [Nilim and Ghaoui, 2005, Wiesemann et al., 2013, Goyal and Grand-Clement, 2020, Rigter et al., 2021]. The solution of a uMDP is a *robust policy* that allows an adversarial selection (i.e., the worst-case scenario) of probabilities within the uncertainty set, and induces a *worst-case performance* (a conservative bound on, e.g., the reachability probability or expected reward). The problem of computing such robust policies, also called *robust verification*, is solved using value iteration or convex optimization, where the uncertainty sets are convex [Wolff et al., 2012, Puggelli et al., 2013].

**Our approach.** We study the problem of learning an MDP from data. We propose an iterative learning method which uses uMDPs as intermediate models and is able to *adapt to new data* which may be inconsistent with prior assumptions. Furthermore, the method is *task-aware* in the sense that the learning procedure respects temporal logic specifications. In particular, our method learns intervals of probabilities for individual transitions. This Bayesian *anytime-learning approach* employs intervals with linearly updating conjugate priors [Walter and Augustin, 2009], and can iteratively improve upon a uMDP that approximates the true MDP we wish to learn. This method not only decreases the size of each interval, but may also increase it again in case of a so-called *prior-data conflict* where new data suggests the actual probability lies outside the current interval. Consequently, a newly learned interval does not need to be a subset of its prior interval. This property makes our method especially suitable to learn MDPs where the transition probabilities change. Alternatively, we also include *probably approximately correct* (PAC) intervals via Hoeffding's inequality [Hoeffding, 1963], which introduces a correctness guarantee for each transition.

We summarize the key features of our learning method, and what sets it apart from other methods.

- *An anytime approach.* The ability to iteratively update intervals that are not necessarily subsets of each other allows us to design an *anytime-learning approach*. At any time, we may stop the learning and compute a robust policy for the uMDP that the process has yielded thus far, together with the worst-case performance of this policy against a given specification. This performance may not be satisfactory, e.g., the worst-case probability to reach a set of critical states may be below a certain threshold. We continue learning towards a new uMDP that more faithfully captures the true MDP due to the inclusion of further data. Thereby, we ensure that the robust policy gradually gets closer to the optimal policy for the true MDP.

- *Specification-driven.* Our method features the possibility to learn in a task-aware fashion, that is, to learn transitions that matter for a given specification. In particular, for reachability or expected reward (temporal logic) specifications which require a certain set of target states to be reached, we only learn and update transitions along paths towards these states. Transitions outside those paths do not affect the satisfaction of the specification.

- *Adaptive to changing environment dynamics.* When using linearly updating intervals, our approach is adaptive to changing environment dynamics. That is, if during the learning process the probability distributions of the underlying MDP change, our method can easily adapt and learns these new distributions.

## 2 Related Work

Uncertain MDPs have (often implicitly) been used by *reinforcement learning* (RL) algorithms, for instance, to optimize the *exploration/exploitation* trade-off by guiding the RL agent towards unexplored parts of the environment, following the *optimism in the face of uncertainty* principle [Jaksch et al., 2010, Fruit et al., 2017]. We use the same principle in the exploration phase of our procedure, but compute robust policies as output to account for the uncertainty in an adversarial (or pessimistic) way. Similarly, uncertain MDPs are used to compute robust policies when the data available is limited [Nilim and Ghaoui, 2005, Wiesemann et al., 2013, Russel and Petrik, 2019]. Such robustness is connected to a *pessimistic* principle that has been effective in offline RL settings [Lange et al., 2012], where the agent only has access to a fixed dataset of past trajectories, meaning it needs to base

decisions on limited data [Rashidinejad et al., 2021, Buckman et al., 2021, Jin et al., 2021]. Likewise, our method may be stopped early and return a policy before the problem is fully explored.

*Robust RL* concerns the standard RL problem, but explicitly accounts for input disturbances and model errors [Morimoto and Doya, 2005]. Often there is a focus on ensuring that a reasonable performance is achieved during data collection. To that end, Lim et al. [2013] and Derman et al. [2019] sample trajectories using a robust policy, which can slow down the process to find an optimal policy. We assume sampling access to the underlying MDP, which allows us to use more efficient exploration. It should be noted that we only use uMDPs as an intermediate model towards learning a standard MDP, whereas in robust RL the model itself may also be an (adversarial) uncertain MDP. As a result, our method converges to an MDP, while robust RL attempts to learn and possibly converge to an uncertain MDP.

Learning MDPs from data is also related to *model learning*, which typically assumes no knowledge about the states and thus iteratively increases the set of states [Vaandrager, 2017]. In [Tappler et al., 2019, 2021] the $L^*$ algorithm for learning finite automata is adapted for MDPs. These methods only yield point estimates of probabilities, and make strong assumptions on the structure of the MDP that is being learned. Ashok et al. [2019] use PAC-learning to estimate the transition functions of MDPs and stochastic games in order to perform *statistical model checking* with a PAC guarantee on the resulting value. In [Bacci et al., 2021] the Baum-Welch algorithm for learning hidden Markov models is adapted to learn MDPs.

Finally, the literature distinguishes two types of uncertainty: *aleatoric* and *epistemic* uncertainty [Hüllermeier and Waegeman, 2021]. Aleatoric uncertainty refers to the uncertainty generated by a probability distribution, like the transition function of an MDP, and is also known as *irreducible uncertainty*. In contrast, epistemic uncertainty is *reducible* by collecting and accounting for (new) data. Our work can be seen as adding an additional layer of epistemic uncertainty on the probability distributions of the transition function that is then, by gathering and including more data, reduced.

## 3 Preliminaries

A *discrete probability distribution* over a finite set $X$ is a function $\mu \colon X \to [0,1] \subset \mathbb{R}$ with $\sum_{x \in X} \mu(x) = 1$. We write $\mathcal{D}(X)$ for the set of all discrete probability distributions over $X$, and by $|X|$ we denote the number of elements in $X$. For any closed interval $I \subseteq \mathbb{R}$ we write $\underline{I}$ and $\overline{I}$ for the lower and upper bounds of the interval, that is, $I = [\underline{I}, \overline{I}]$.

**Definition 1** (Markov decision process). *A Markov decision process (MDP) is a tuple $(S, s_I, A, P, R)$ with $S$ a finite set of states, $s_I \in S$ the initial state, $A$ a finite set of actions, $P \colon S \times A \times S \to [0,1]$ with $\forall s, a \in S \times A, \sum_{s'} P(s, a, s') = 1$ (such that $P(s, a) \in \mathcal{D}(S)$) the probabilistic transition function, and $R \colon S \times A \to \mathbb{R}_{\geq 0}$ the reward (or cost) function.*

A *trajectory* in an MDP is a finite sequence $(s_0, a_0, s_1, a_1, \ldots, s_n) \in (S \times A)^* \times S$ where $s_0 = s_I$ and $P(s_i, a_i, s_{i+1}) > 0$ for $0 \leq i < n$. A *memoryless* policy is a function $\pi \colon S \to \mathcal{D}(A)$. If $\pi$ maps to Dirac distributions, then it is a memoryless *deterministic* (or pure) policy. Applying a policy to an MDP $M$ resolves all the non-deterministic choices and yields an (induced) *discrete-time Markov chain* (DTMC); see, e.g., Puterman [1994] for details.

**Definition 2** (Uncertain MDP). *An uncertain MDP (uMDP) is a tuple $(S, s_I, A, \mathbb{I}, \mathcal{P}, R)$ where $S, s_I, A, R$ are as for MDPs, $\mathbb{I}$ is a set of probability intervals $\mathbb{I} = \{[a, b] \mid 0 < a \leq b \leq 1\}$, and $\mathcal{P} \colon S \times A \times S \to (\mathbb{I} \cup \{0\})$ is the uncertain transition function, assigning either a probability interval, or the exact probability $0$ to any transition.*

Uncertain MDPs can be seen as an uncountable set of MDPs that only differ in their transition functions. For a transition function $P$, we write $P \in \mathcal{P}$ if for every transition the probability of $P$ lies within the interval of $\mathcal{P}$, i.e., $P(s, a, s') \in \mathcal{P}(s, a, s')$ for all $(s, a, s') \in S \times A \times S$. We only allow intervals with a lower bound greater than zero, to ensure a transition cannot vanish under certain distributions generated by the uncertainty. This assumption is standard in uMDPs [Wiesemann et al., 2013, Puggelli et al., 2013]. Applying a policy to a uMDP yields an induced interval Markov chain [Jonsson and Larsen, 1991].

**Specifications.** We consider *reachability* or *expected reward (cost)* specifications. The value $\mathbb{P}_\pi^M(\lozenge T)$ is the probability to reach a set of *target* states $T \subseteq S$ in the MDP $M$ under the policy $\pi$,

also referred to as the *performance of* $\pi$.[2] Likewise, $R^M_\pi(\Diamond T)$ describes the expected accumulated reward to reach $T$ under $\pi$. Note that for probabilistic specifications, we can replace the formula $\Diamond T$ by more general reach-avoid specifications using the until operator from temporal logic.

Formally, the specification $\mathbb{P}_{\text{Max}}(\Diamond T) = \max_\pi \mathbb{P}_\pi(\Diamond T)$ expresses that the probability of eventually reaching the target set $T \subseteq S$ should be maximized. Likewise, the specification $R_{\text{Max}}(\Diamond T)$ requires the expected reward for reaching $T$ to be maximized. For minimization, we write $\mathbb{P}_{\text{Min}}(\Diamond T)$ and $R_{\text{Min}}(\Diamond T)$, respectively. Besides optimizing a probability or reward, a specification may also express an explicit user-provided threshold to compare the performance of a policy to.

For uMDPs, we define *optimistic* and *pessimistic* specifications. In optimistic specifications, we assume the best-case scenario of the uncertainty to satisfy the specification by also minimizing (or maximizing) over the uncertainty set, written as $\mathbb{P}_{\text{MinMin}}(\Diamond T) = \min_\pi \min_{P \in \mathcal{P}} \mathbb{P}^{\mathcal{M}[P]}_\pi(\Diamond T)$ (or $\mathbb{P}_{\text{MaxMax}}(\Diamond T)$), where the first Min (Max) signals what the decision-maker is trying to achieve, and the second what the uncertainty does. In pessimistic specifications, the uncertainty does the opposite of the goal: $\mathbb{P}_{\text{MaxMin}}(\Diamond T)$ or $\mathbb{P}_{\text{MinMax}}(\Diamond T)$. The notation is similar for reward specifications. A (standard) specification $\varphi$ can be extended to be optimistic or pessimistic by adding a second Min or Max. We write $\varphi_O$ for its optimistic extension, and $\varphi_P$ for its pessimistic extension.

For an MDP $M$, the aim is to compute a policy $\pi$ that either optimizes a given specification $\varphi$, or whose performance respects a given threshold that, e.g., provides an upper bound on the probability of reaching a set of critical states. Common methods are value iteration or linear programming [Puterman, 1994, Baier and Katoen, 2008]. For uncertain MDPs $\mathcal{M}$, the goal is to compute a policy $\pi$ that satisfies an optimistic or pessimistic specification $\varphi_O$ or $\varphi_P$. In the latter case, we call $\pi$ a *robust policy*. Optimal policies in uMDPs can be computed via robust dynamic programming or convex optimization [Wolff et al., 2012, Puggelli et al., 2013].

## 4  Problem Statement and Outline of the Procedure

We have an unknown but fixed MDP $M = (S, s_I, A, P, R)$, which we will refer to as the *true MDP*, an *initial prior uMDP* $\mathcal{M} = (S, s_I, A, \mathbb{I}, \mathcal{P}, R)$, and a specification $\varphi$ which we want to satisfy. A discussion of prior (and other parameter) choices follows in Section 6.

**Assumption 1** (Underlying graph). *We assume that the underlying graph of the true MDP is known. In particular: transitions that do not exist in the true MDP (transitions of probability* 0*) do also not exist in the uMDP, transitions of probability* 1 *in the true MDP are assigned the point interval* $[1,1]$ *in the uMDP, and any other transition of non-zero probability* $p$ *has an interval* $I \in \mathbb{I}$ *in the uMDP.*

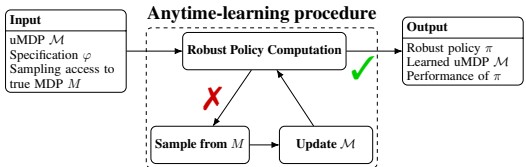

Figure 1: Procedure outline.

Under Assumption 1 and Definition 2, we construct the initial prior uMDP $\mathcal{M}$ to have transitions of probability 0 and 1 exactly where the true MDP $M$ has these too, and interval transitions $[\varepsilon, 1 - \varepsilon]$ for all other transitions, with $\varepsilon > 0$ free to choose. In particular, our approach does not require $\varepsilon$ to be smaller than the smallest probability $p > 0$ occurring in $M$, which we also do not assume to be known. Alternatively, in case further knowledge is available, one may use any other prior uMDP as long as it satisfies Assumption 1. Our learning problem is expressed as follows:

> The problem is to learn the transition probabilities of a true MDP $M$, driven by a specification $\varphi$, via intermediate uncertain MDPs $\mathcal{M}$ that are iteratively updated to account for newly collected data, such that at any time a robust policy can be computed.

In the following we outline our anytime-learning procedure as illustrated in Figure 1.

1. **Input.** We start with an initial prior uMDP $\mathcal{M}$ and a specification $\varphi$ we wish to verify. We assume (black box) access to the unknown *true MDP* $M$ to sample trajectories from, or alternatively, assume a (constant) stream of observations from the true MDP.

---

[2]We will omit the policy $\pi$ and the MDP $M$ whenever they are clear from the context.

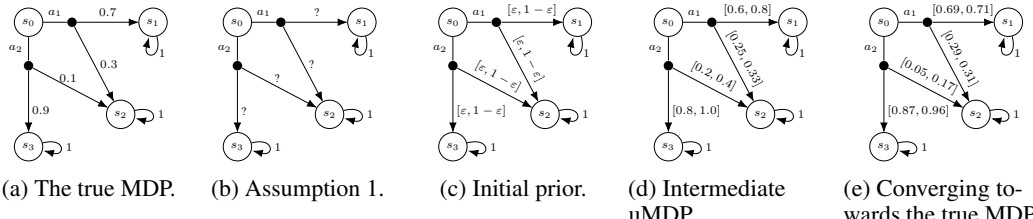

(a) The true MDP.    (b) Assumption 1.    (c) Initial prior.    (d) Intermediate uMDP.    (e) Converging towards the true MDP.

Figure 2: Process flow on an example MDP.

2. **Robust policy computation.** We compute a robust policy $\pi$ for the pessimistic extension of $\varphi$, i.e. $\varphi_P$, in the uncertain MDP $\mathcal{M}$, together with the worst-case performance of the specification in the current uMDP. If the specification contains an explicit threshold, the value can be compared against the threshold for automatic termination. In case of specifications that optimize a probability or reward, termination needs to be done manually, as it is impossible to tell if the maximum (minimum) was achieved.

3. **Anytime learning.** If the result from step 2 is unsatisfactory, we start learning:

   (a) **Exploration.** We sample one or more trajectories from the true MDP $M$, using the *optimism in the face of uncertainty* principle.

   (b) **Update.** We update the intervals of the uMDP $\mathcal{M}$ in accordance with the newly collected data. This update yields a new uMDP that more faithfully captures all collected data up to this point than the previous uMDP.

   (c) **Repeat.** We start again at step 2 with this new uMDP until (manual) termination.

4. **Output.** The process may be stopped at any moment and yields the latest uMDP $\mathcal{M}$ together with robust policy $\pi$ and the performance of $\pi$ on $\mathcal{M}$.

The effects of this procedure are illustrated in Figure 2. In 2a, we see an example MDP $M$ to learn, and 2b shows the assumed knowledge about $M$. 2c shows the initial uMDP $\mathcal{M}$ constructed from 2b, using a (symbolic or explicit) lower bound $\varepsilon > 0$ to ensure that all lower bounds of $\mathcal{M}$ are strictly greater than zero. In 2d, we see an intermediate learned uMDP. Some intervals may already have successfully converged towards the probability of that transition in the true MDP, while others may be very inaccurate due to a low sample size and thus a bad estimate. Finally, 2e depicts the learned uMDP converging towards the true MDP.

## 5 Bayesian Learning

We assume access to trajectories through the true MDP. Due to the Markov property of the MDP, i.e., the fact that the transition probabilities only depend on the current state and not on any further history, we may split each trajectory $\tau$ into separate sets of independent experiments where a state-action pair $(s, a)$ is sampled and a successor state $s_i$ is observed; see also Appendix A of [Strehl and Littman, 2008]. We count the number of occurrences of the transition $(s, a, s_i)$ in each trajectory $\tau$, and the number of occurrences of the state-action pair $(s, a)$ in each trajectory $\tau$, denoted by $\#(s, a, s_i)$ and $\#(s, a)$, respectively. In the following, we introduce the two approaches of learning intervals: the standard approach of PAC learning via Hoeffding's inequality which provides a PAC guarantee on the value of the learned model, and a new approach using *linearly updating intervals* (LUI), which is more flexible due to the inclusion of *prior-data conflicts* and its closure properties. Finally, we introduce a minor modification to the LUI approach, making it also applicable to situations where the underlying MDP changes over time. Such a modification is not possible in PAC learning as Hoeffding's inequality assumes independent samples from a fixed distribution.

### 5.1 Learning PAC Intervals

We use the standard method of maximum a-posteriori probability (MAP) estimation to infer point estimates of probabilities; see Appendix A for details. These point estimates can then easily be turned into *probably approximately correct* (PAC) intervals via Hoeffding's inequality. Given $N = \#(s, a)$ samples and a fixed *error rate* $\gamma$, we use Hoeffding's inequality [Hoeffding, 1963] to compute the

interval size $\delta = \sqrt{\log(2/\gamma)/2N}$. This $\delta$ may then be used to construct intervals that are PAC with respect to the true transition probability. To lift the PAC guarantee to the entire learned MDP, and consequently also the optimal value for some specification, we need to distribute $\delta$ over all transitions, i.e., $\delta_P = \delta / \sum_{(s,a) \in S \times A} |\mathbf{Succ}(s,a)|$, where $\mathbf{Succ}(s,a)$ is the set of successor states of $(s,a)$. Using this $\delta_P$, we then construct the intervals

$$\mathcal{P}(s, a, s_i) = [\max(\varepsilon, \tilde{P}(s, a, s_i) - \delta_P), \min(\tilde{P}(s, a, s_i) + \delta_P, 1)], \tag{1}$$

where $\tilde{P}$ is the (MAP) point estimate, and $\varepsilon$ is again a small value to ensure that the interval lower bounds are non-zero. As a result, we have the following Proposition, whose proof is a direct application of Hoeffding's inequality.

**Proposition 1.** *The true MDP $M^*$ lies within the learned uMDP $\mathcal{M}$ with probability greater or equal to $1 - \gamma$.*

This is a standard result and also used in, e.g., PAC statistical model checking [Ashok et al., 2019].

### 5.2 Learning Linearly Updating Probability Intervals

We use the Bayesian approach of *intervals with linearly updating conjugate priors* [Walter and Augustin, 2009] to learn intervals of probabilities. Each uncertain transition $\mathcal{P}(s, a, s_i)$ is assigned a *prior interval* $\mathcal{P}_i = [\underline{\mathcal{P}}_i, \overline{\mathcal{P}}_i]$, and a *prior strength interval* $[\underline{n}_i, \overline{n}_i]$ that represents a minimum and maximum number of samples on which the prior interval is based. The greater the values of the strength interval, the more emphasis is placed on the prior, and the more data is needed to significantly change the prior when computing the posterior. The greater the difference between the $\underline{n}_i$ and $\overline{n}_i$, the greater the difference between a *prior-data conflict* and a *prior-data agreement*.

**Definition 3** (Posterior interval computation). *The interval $[\underline{\mathcal{P}}_i, \overline{\mathcal{P}}_i]$ can be updated to $[\underline{\mathcal{P}}'_i, \overline{\mathcal{P}}'_i]$, using $N = \#(s,a)$ and $k_i = \#(s, a, s_i)$, as follows:*

$$\underline{\mathcal{P}}'_i = \begin{cases} \frac{\overline{n}_i \underline{\mathcal{P}}_i + k_i}{\overline{n}_i + N} & \text{if } \forall j. \frac{k_j}{N} \geq \underline{\mathcal{P}}_j \text{ (prior-data agreement)}, \\ \frac{\underline{n}_i \underline{\mathcal{P}}_i + k_i}{\underline{n}_i + N} & \text{if } \exists j. \frac{k_j}{N} < \underline{\mathcal{P}}_j \text{ (prior-data conflict)}. \end{cases} \tag{2}$$

$$\overline{\mathcal{P}}'_i = \begin{cases} \frac{\overline{n}_i \overline{\mathcal{P}}_i + k_i}{\overline{n}_i + N} & \text{if } \forall j. \frac{k_j}{N} \leq \overline{\mathcal{P}}_j \text{ (prior-data agreement)}, \\ \frac{\underline{n}_i \overline{\mathcal{P}}_i + k_i}{\underline{n}_i + N} & \text{if } \exists j. \frac{k_j}{N} > \overline{\mathcal{P}}_j \text{ (prior-data conflict)}. \end{cases} \tag{3}$$

*The strength interval is updated by adding the number of samples $N$ to it: $[\underline{n}'_i, \overline{n}'_i] = [\underline{n}_i + N, \overline{n}_i + N]$.*

The initial values for the priors of each state-action pair can be chosen freely subject to the constraints $0 < \underline{\mathcal{P}}_i \leq \overline{\mathcal{P}}_i \leq 1$, and $\overline{n}_i \geq \underline{n}_i \geq 1$.

**Key properties of linearly updating probability intervals.**

- *Convergence in the infinite run.* Under the assumption that the true MDP does not change, each interval will converge to the exact transition probability when the total number of samples processed tends to infinity, regardless of how many samples are processed *per iteration* [Walter and Augustin, 2009]. This assumption is, however, not required for our work. If the true MDP changes over time, or is *adversarial* (i.e., a uMDP), our method is still applicable, but will not converge to a fixed MDP.

- *Prior-data conflict.* When the estimated probability $k_i/N$ lies outside the current interval, a so-called *prior data conflict* occurs. Consequently, if at some point we derive an interval that does not contain the true transition probability, the method will correct itself later on.

- *Closure properties under updating.* Finally, updating is closed in two specific ways. First, any interval of probabilities is updated again to a valid interval of probabilities, and second, any set of intervals at a state-action pair that contains a valid distribution over the successor states will again contain a distribution over successor states after updating.

A key requirement for computing robust policies on uMDPs is that the lower bound of every interval is strictly greater than zero (see Definition 2). This closure property is formalized as follows.

**Theorem 1** (Closure of intervals under learning). *For any valid prior interval $[\underline{\mathcal{P}}, \overline{\mathcal{P}}]$ with $0 < \underline{\mathcal{P}} \leq \overline{\mathcal{P}} \leq 1$, we have that the posterior $[\underline{\mathcal{P}}', \overline{\mathcal{P}}']$ computed via Definition 3 also satisfies $0 < \underline{\mathcal{P}}' \leq \overline{\mathcal{P}}' \leq 1$.*

Furthermore, we also have closure properties at each state-action pair. In particular, if we choose our prior intervals such that there is at least one valid probability distribution at the state-action pair, then the posterior intervals will again contain a valid probability distribution. We formalize this notion by examining the sum of the lower and upper bounds of the intervals in the following Theorem:

**Theorem 2** (Closure of distributions under learning). *We have the following bounds on the set of posterior distributions. In case of a prior data agreement, we have that the sum of the posterior bounds is bounded by the sum of the prior bounds, and the value $1$. That is,*

$$\sum_i \underline{\mathcal{P}}_i \leq \sum_i \underline{\mathcal{P}}'_i \leq 1, \qquad 1 \leq \sum_i \overline{\mathcal{P}}'_i \leq \sum_i \overline{\mathcal{P}}_i. \tag{4}$$

*In case of a prior-data conflict, the sum of posterior bounds is no longer necessarily bounded by the prior, and we only have*

$$\sum_i \underline{\mathcal{P}}'_i \leq 1 \leq \sum_i \overline{\mathcal{P}}'_i. \tag{5}$$

*Note, however, that this last constraint* (5) *is already sufficient to ensure that there is a valid probability distribution at the state-action pair.*

The proofs for both Theorem 1 and 2 can be found in Appendix B.

### 5.3 Efficient exploration

Above, we assumed that a set of trajectories was given. To actually obtain the trajectories, we use the well-established *optimism in the face of uncertainty* principle [Munos, 2014]. We compute the optimal policy for the optimistic extension (see Section 3) of the specification $\varphi$, i.e. $\varphi_O$, in the current uMDP and use this policy for exploration. To make exploration specification-driven, we only sample transitions along trajectories towards the target state(s) of the specification. When the last seen state has probability zero or one to reach the target for reachability, or reward zero or infinity, we restart. States satisfying these conditions can be found by analyzing the graph of the true MDP [Baier and Katoen, 2008].

**Trajectories and iterations.** As explained in Section 4, our method is iterative in terms of updating the uMDP $\mathcal{M}$ and computing a robust policy. After updating the uMDP, we also compute a new exploration policy, based on the new uMDP. Each iteration consists of processing at least one, but possibly more trajectories. To determine how many trajectories to collect, we use a doubling-counting scheme, where we keep track of how often every state-action pair and transition is visited during exploration [Jin et al., 2020]. An iteration is completed when any of the counters is doubled with respect to the previous iteration. A detailed description of this schedule is given in Appendix C.

### 5.4 Learning under changing environments

Previously, we assumed a fixed unknown true MDP $M^*$ to learn. But what if the transition probabilities of $M^*$ suddenly change? More precisely, we assume two unknown true MDPs, $M_1^*$ and $M_2^*$ with the same underlying graph but (possibly) different transition probabilities. After an unknown number of interactions with the true environment $M_1^*$, we suddenly continue interacting with $M_2^*$. We modify our LUI approach by introducing a bound on the strength of the prior, $n_{\text{Max}} = [\underline{n}_{\text{Max}}, \overline{n}_{\text{Max}}]$, and update the strength intervals by the following rule

$$[\underline{n}'_i, \overline{n}'_i] = [\min(\underline{n}_i + N, \underline{n}_{\text{Max}}), \min(\overline{n}_i + N, \overline{n}_{\text{Max}})].$$

The probability intervals themselves are updated in the same way as before in Definition 3. By limiting the prior strength in this way, we trade some rate of convergence for adaptability. The weaker the prior, the greater the effect of a prior-data conflict, and hence adaptability to new data. When suddenly changing environments, new data will likely lead to prior-data conflicts, and thus a higher adaptability of the overall learning method.

We add randomization to the pure optimistic exploration policy. We introduce a hyperparameter $\xi \in [0, 1]$, and follow with probability $\xi$ the action of the optimistic policy, and distribute the remaining $1 - \xi$ uniformly over the other actions, yielding a memoryless randomized policy.

# 6 Experimental Evaluation

We implement our approach, with both linearly updating intervals (LUI) and PAC intervals (PAC), in Java on top of the verification tool PRISM [Kwiatkowska et al., 2011], together with a variant of value iteration to compute robust policies for uMDPs with convex uncertainties [Wolff et al., 2012].[3] We compare our method to point estimates derived via MAP-estimation (MAP) and with uMDPs learned by the UCRL2 reinforcement learning algorithm [Jaksch et al., 2010] (UCRL2). We modify UCRL2 to make it more comparable to our setting. In particular, we use optimistic policies for exploration, but robust policies to compute the performance, in contrast to the standard UCRL2 setting which only uses optimistic policies, see Appendix E for further details.

Without knowledge about the true MDP apart from Assumption 1, we have to define an appropriate prior interval for every transition. We set $\varepsilon = \texttt{1e-4}$ as constant and define the prior uMDP with intervals $\mathcal{P}_i = [\varepsilon, 1 - \varepsilon]$ and strength intervals $[\underline{n}_i, \overline{n}_i] = [5, 10]$ at every transition $\mathcal{P}(s, a, s_i)$, as in Figure 2c. For MAP, we use a prior of $\alpha_i = 10$ for all $i$. The same prior is used for the point estimates of both PAC and UCRL2, together with an error rate of $\gamma = 0.01$.

**Evaluation metrics.** We consider two metrics to evaluate the four learning methods.

- *Performance.* How does the robust policy computed on the learned model perform on the true MDP? We evaluate the probability of satisfying the given specification $\mathbb{P}^M_\pi(\lozenge T)$ or expected reward $\mathrm{R}^M_\pi(\lozenge T)$ of the robust policy $\pi$ computed after each update of the model.

- *Performance estimation error.* How well do we *expect* a robust policy to perform on the true MDP based on the performance on the intermediate learned uMDP? We compute the difference between the performance of the robust policy on the learned uMDP (the worst-case performance) and the performance on the true MDP. While values closer to zero are preferable, we do not accept methods with positive estimation errors, since this indicates their estimated performance is not a lower (conservative) bound on the actual performance of the policy. In particular, an estimation error above zero shows the policy is misleading in terms of predicting its performance.

We benchmark our method using several well-known environments: the Chain Problem [Araya-López et al., 2011], Aircraft Collision Avoidance [Kochenderfer, 2015], a slippery Grid World [Derman et al., 2019], a 99-armed Bandit [Lattimore and Szepesvári, 2020], and two version of a Betting Game [Bäuerle and Ott, 2011]. For details on all these environments we refer to Appendix D. We highlight the Betting Game and Chain Problem environments here, as they will be used to explain some of the key observations we make from our experimental results.

- *Betting Game.* The agent starts with 10 coins and attempts to maximize the number of coins after 6 bets. When a bet is won, the number of coins placed is doubled; when lost, the number of coins placed is removed. The agent may bet 0, 1, 2, 5, or 10 coins. We consider two versions of the game, one which is favourable to the player, with a win probability of 0.8, and one that is unfavourable with win probability 0.2. After 6 bets the player receives a reward equal to the number of coins left. The specification is to maximize the reward.

- *Chain Problem.* We consider a chain of 30 states. There are three actions, one progresses with probability 0.95 to the next state, and resets the model to the initial state with probability 0.05. The second action does the same, but with reversed probabilities. The third action has probability 0.5 for both cases. Every action gets a reward of 1. As specification, we minimize the reward to reach the last state of the chain.

## Results

We present an excerpt of our experimental results here, and refer the reader to Appendix F for the full set of results, which in particular also includes the estimation error for all environments, an additional model error metric, and further experiments regarding different priors and changing environments. All experiments were performed on a machine with a 4GHz Intel Core i9 CPU, using a single core. Each experiment is repeated 100 times, and reported with a 95% confidence interval.

---

[3]The implementation is available at `https://github.com/LAVA-LAB/luiaard`.

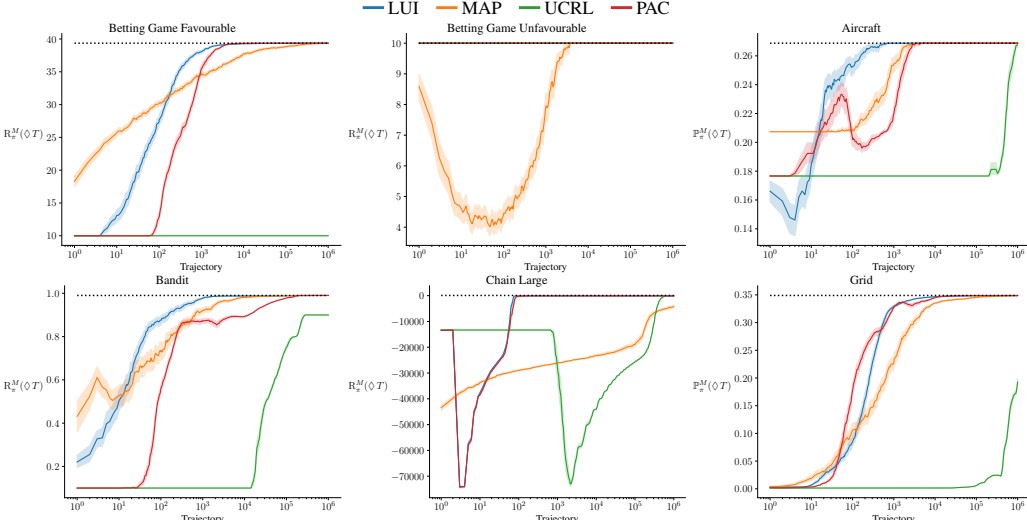

Figure 3: Comparison of the performance of robust policies on different environments against the number of trajectories processed (on log-scale). The dashed line indicates the optimal performance.

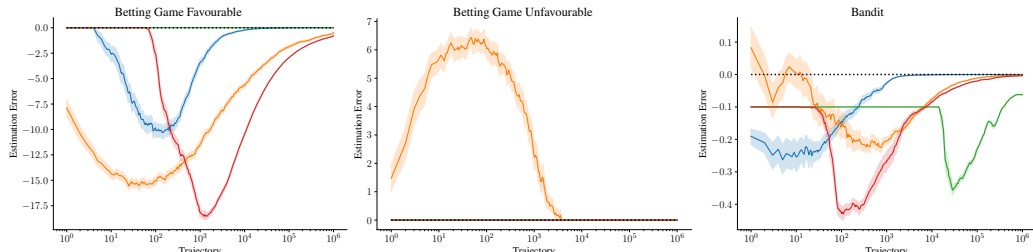

Figure 4: Estimation error on the two Betting Game environments and the Bandit, against the number of trajectories processed (on log-scale).

Figure 3 shows the performance of the robust policies computed via each learning method against the number of trajectories processed on the different environments. We first note that in all environments, our LUI method is the first to find an optimal policy. Depending on the environment, the performance of LUI and PAC may be roughly equivalent (Chain, Grid). When also taking the estimation error into account (Figure 4), we see LUI outperforming other methods on that metric. UCRL2 is the slowest to converge to an optimal policy. This due to UCRL2 being a reinforcement learning algorithm, and thus it is slower in reducing the intervals in favour of a broader exploration.

**Recovering from bad estimates.** On the Chain environment, we see LUI and PAC (around trajectory 5), and UCRL2 (around trajectory $10^3$) choose the wrong action(s), with an decrease in performance as a result. This is most likely due to getting a bad estimate on some of the transitions later on in the chain, leading to many resets to the initial state, and thus decreasing the performance significantly. While all three methods manage to recover and then find the optimal policy, UCRL2 takes significantly longer: only after trajectory $10^5$, where LUI and PAC only need about 100 trajectories. MAP-estimation typically sits between LUI and PAC in terms of performance. It is less sensitive to mistakes like the one discussed above, but is less reliable in providing a conservative bound on its performance, as will be discussed below. Furthermore, we see that in the unfavourable Betting Game, only MAP-estimation gives sub-optimal performance, due to bad estimates. It is able to recover from this, but needs almost $10^4$ trajectories to do so. Due to the low win probability in this Betting Game, a robust policy on the uMDPs is by default an optimal policy for the true MDP, and we see that LUI, PAC, and UCRL2 do not change to a sub-optimal policy.

**Robust policies are conservative.** Consider Figure 4. We note the undesirable behaviour of having an Estimation Error above zero, which means the performance of the policy on the learned model

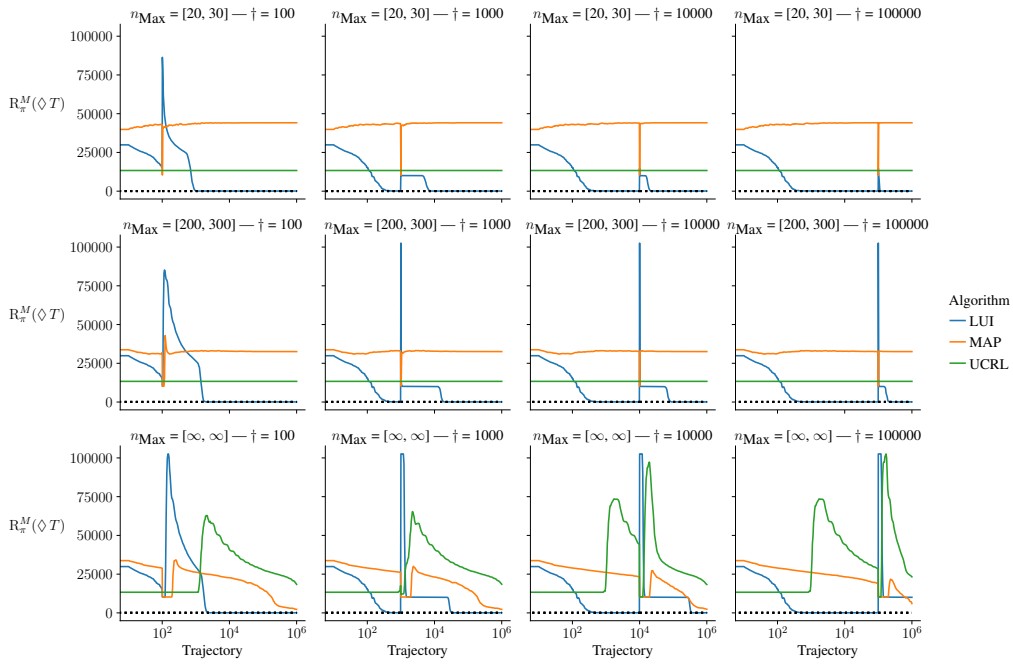

Figure 5: Environment change on the Chain Problem at different points $\dagger \in \{10^2, \ldots, 10^5\}$ using randomization parameter $\xi = 0.8$ in the exploration.

was higher than the performance of that policy on the true MDP. MAP-estimation is particularly susceptible to this, while all three uMDP methods yield policies that are conservative in general, though some exceptions exist as shown in the full results in Appendix F.

**LUI is adaptive to changing environments.** Finally, we investigate the behaviour of the learning methods when after a fixed number of trajectories the probabilities of the true MDP change, as introduced in Section 5.4. Figure 5 shows the performance of the robust policy for each learning method on the Chain environment, results for the same experiment on the Betting Game can be found in Appendix F.After $\dagger \in \{10^2, \ldots, 10^5\}$ trajectories, we change the environment by swapping the transition probabilities, for three different bounds on the strength intervals. We similarly bound the MAP-estimation priors for MAP and UCRL. PAC is omitted from this experiment as PAC guarantees lose all meaning when changing the underlying distribution. After the change in environment, the new optimal policy has to use the opposite action from the previously optimal policy. We see that LUI is the only method capable of converging to optimal policies both before and after the change in environment. Furthermore, the lower the bounds on the prior strength, the faster it adapts to the change. We conclude that LUI is an effective approach to deal with learning an MDP under potential adversarial sampling conditions.

## 7 Conclusion and Future Work

We presented a new Bayesian method that learns uMDPs to approximate an MDP, either via linearly updating intervals, or PAC-intervals. Robust policies computed on learned uMDPs are shown to be conservative and reliable in predicting their performance when applied on the MDP that is being learned. The approach of linearly updating intervals is also effective at continuously learning a potentially changing environment. For future work, we aim to extend our method to uncertain POMDPs [Suilen et al., 2020, Cubuktepe et al., 2021]. While we do not see any immediate negative societal impacts of our work, we acknowledge that potential misuse of our work cannot be ruled out due to the generality of MDPs.

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
