# Robust Anytime Learning of Markov Decision Processes (Supplementary Material) [*]

**Marnix Suilen**
Department of Software Science
Radboud University
Nijmegen, The Netherlands

**Thiago D. Simão**
Department of Software Science
Radboud University
Nijmegen, The Netherlands

**David Parker**
Department of Computer Science
University of Oxford
Oxford, United Kingdom

**Nils Jansen**
Department of Software Science
Radboud University
Nijmegen, The Netherlands

## A    MAP-estimation

We describe a general setup for learning point estimates of probabilities via *maximum a-posteriori estimation* (MAP-estimation). For a fixed state-action pair $(s, a)$ with $m$ different successor states $s_1, \ldots, s_m$ that is sampled $N = \#(s, a)$ times, and where each successor state is observed $k_i = \#(s, a, s_i)$ times for $i = 1, \ldots, m$, we have the *multinomial likelihood*

$$Mn(k_1, \ldots, k_n \mid P(s, a, \cdot)) = \frac{N!}{k_1! \cdot \ldots \cdot k_m!} \cdot \prod_{i=1}^{m} P(s, a, s_i)^{k_i}, \tag{1}$$

where the transition probabilities $P(s, a, s_i)$ are unknown and given by a Dirichlet distribution with parameters $\alpha_1, \ldots, \alpha_m$ [Bishop, 2007]:

$$Dir(P(s, a, \cdot) \mid \alpha_1, \ldots, \alpha_m) \propto \prod_{i=1}^{m} P(s, a, s_i)^{\alpha_i - 1}. \tag{2}$$

The Dirichlet distribution is a *conjugate prior* to the multinomial likelihood, meaning that we do not need to compute the posterior distribution explicitly, but instead we just update the parameters of the prior Dirichlet distribution. Formally, conjugacy is a closure property, see [Jacobs, 2020].

Given a prior Dirichlet distribution with parameters $\alpha_1, \ldots, \alpha_m$, and observing the $i$-th successor state $k_i$ times, the posterior Dirichlet distribution is given by simply adding $k_i$ to parameter $\alpha_i$:

$$Dir(P(s, a, \cdot) \mid \alpha_1 + k_1, \ldots, \alpha_m + k_m). \tag{3}$$

Having computed the posterior Dirichlet distribution, MAP-estimation can be used to infer the probabilities. These estimates are given by the *mode* of each parameter of the posterior distribution:

$$\tilde{P}(s, a, s_i) = \frac{\alpha_i - 1}{(\sum_{j=1}^{m} \alpha_j) - m}. \tag{4}$$

When all parameters $\alpha_i$ are equal, MAP-estimation yields uniform distributions.

---

[*]Supported by NWO OCENW.KLEIN.187: "Provably Correct Policies for Uncertain Partially Observable Markov Decision Processes", NWA.1160.18.238 (PrimaVera) and by the ERC under the European Union's Horizon 2020 research and innovation programme (FUN2MODEL, grant agreement No. 834115).

36th Conference on Neural Information Processing Systems (NeurIPS 2022).

# B Proofs

The proof of Theorem 1 is straightforward and relies on the observation that for any amount of finite data, a single update can only grow closer to zero but not become zero, and that iterated updates converge to the true probability which is assumed to be non-zero.

*Proof Theorem 1.* For any transition $(s, a, s_i)$ the following holds. We assume a valid prior, that is, $0 < \underline{\mathcal{P}}_i \leq \overline{\mathcal{P}}_i \leq 1$. We have an empirical estimate of $k_i/N$. Then we also have $0 < \underline{\mathcal{P}}'_i \leq \overline{\mathcal{P}}'_i \leq 1$, where the first inequality $0 < \underline{\mathcal{P}}'_i$ follows from the fact that Equation 2 can only be 0 when their nominator is zero, which is not possible when $\underline{n}_i \geq 1$ (or $\overline{n}_i \geq 1$) and $\underline{\mathcal{P}}_i > 0$. The second inequality, $\underline{\mathcal{P}}'_i \leq \overline{\mathcal{P}}'_i$ follows directly from Equations 2 and 3 together with $\underline{\mathcal{P}}_i \leq \overline{\mathcal{P}}_i$. The third inequality $\overline{\mathcal{P}}'_i \leq 1$ follows again from Equation 3 when $\overline{\mathcal{P}}_i \leq\leq 1$ and $k_i \leq N$. All reasoning above is independent of prior-data agreement or conflict and thus applies to both cases in each of the equations. □

The proof of Theorem 2 uses the following Lemma:

**Lemma 1.** *At a state-action pair with $m$ successor states, there can be at most $m - 1$ prior-data conflicts on the upper (or lower) bound.*

*Proof.* We prove the lemma for the upper bound, a proof for the lower bound follows by symmetry. Assume a valid prior, that is, $\sum_i \overline{\mathcal{P}}_i \geq 1$. Suppose there is a prior-data conflict for every interval, i.e., $\forall i. \frac{k_i}{N} > \overline{\mathcal{P}}_i$. Then we also have

$$1 = \sum_i \frac{k_i}{N} > \sum_i \overline{\mathcal{P}}_i \geq 1,$$

which is clearly not possible. The possibility for $m - 1$ prior-data conflicts is witnessed in the following example. Take a state-action pair with two successor states, $s_1$ and $s_2$. Then $m = 2$. Take one sample, i.e., $N = 1$, and suppose we observe $s_1$, such that $k_1 = 1$ and $k_2 = 0$. Then for any valid prior intervals $I_1 = [\underline{\mathcal{P}}_1, \overline{\mathcal{P}}_1]$ and $I_2 = [\underline{\mathcal{P}}_2, \overline{\mathcal{P}}_2]$ we have $k_1/N = 1 > \overline{\mathcal{P}}_1$ and $k_2/N = 0 < \overline{\mathcal{P}}_2$. Hence, a prior-data conflict at $I_1$ but not at $I_2$, thus $m - 1$ conflicts at the state-action pair in total. □

We additionally have the following Lemma:

**Lemma 2.** *A prior-data conflict at the upper bound implies a prior-data agreement at the lower bound, and vice versa.*

*Proof.* Assume a conflict at the upper bound $\overline{\mathcal{P}}_i$. Then $\frac{k_i}{N} > \overline{\mathcal{P}}_i \geq \underline{\mathcal{P}}_i$, which is a prior-data agreement with $\underline{\mathcal{P}}_i$ by definition. The other way around follows by symmetry. □

Now we prove Theorem 2.

*Proof Theorem 2.* We start with the constraints in Equation 4.

First, $1 \leq \sum_i \overline{\mathcal{P}}'_i$. We use that there is a prior-data agreement, that is, $\forall i. \frac{k_i}{N} \leq \overline{\mathcal{P}}_i$. Then we derive

$$\sum_i \overline{\mathcal{P}}'_i = \sum_i \frac{\overline{n}_i \overline{\mathcal{P}}_i + k_i}{\overline{n}_i + N} \quad \geq \quad \sum_i \frac{\overline{n}_i \frac{k_i}{N} + k_i}{\overline{n}_i + N} = \sum_i \frac{\frac{\overline{n}_i k_i + k_i N}{N}}{\overline{n}_i + N}$$

$$= \sum_i \frac{\frac{k_i(\overline{n}_i + N)}{N}}{\overline{n}_i + N} = \sum_i \frac{k_i}{N} = \frac{1}{N} \sum_i k_i = \frac{N}{N} = 1.$$

The bound $\sum_i \overline{\mathcal{P}}'_i \leq \sum_i \overline{\mathcal{P}}_i$ is derived using that

$$\forall i. \frac{k_i}{N} \leq \overline{\mathcal{P}}_i \iff \forall i. k_i \leq \overline{\mathcal{P}}_i N.$$

Then, it follows that

$$\sum_i \overline{\mathcal{P}}'_i = \sum_i \frac{\overline{n}_i \overline{\mathcal{P}}_i + k_i}{\overline{n}_i + N} \quad \leq \quad \sum_i \frac{\overline{n}_i \overline{\mathcal{P}}_i + \overline{\mathcal{P}}_i N}{\overline{n}_i + N} = \sum_i \frac{\overline{\mathcal{P}}_i(\overline{n}_i + N)}{\overline{n}_i + N} = \sum_i \overline{\mathcal{P}}_i.$$

The proof for the bounds

$$\sum_i \underline{\mathcal{P}}_i \leq \sum_i \underline{\mathcal{P}}'_i \leq 1$$

is symmetrical to the one above.

Next, we consider the case for a prior-data conflict, that is, the bounds from Equation 5. The existential condition $\exists j. \frac{k_j}{N} \geq \overline{\mathcal{P}}_j$ does not have to be unique, hence we make a case distinction on the indexes for which the existential quantification holds and for which it does not. Let $I = \{1, \ldots, m\}$ be the set of indices that enumerates the $m$ successor states at the state-action pair we consider. By Lemma 1 we know that there are at most $m - 1$ prior-data conflicts. Hence, we can partition $I$ into two non-empty subsets, $I_A$ containing all indices where the point estimate agrees with the prior interval, and $I_C$ the set of indices where there is a prior-data conflict. That is,

$$I_A = \{i \in I \mid \frac{k_i}{N} \leq \overline{\mathcal{P}}_i\}, \qquad I_C = \{i \in I \mid \frac{k_i}{N} > \overline{\mathcal{P}}_i\}.$$

We use this partition to split the sum over all indices in two:

$$\sum_i \overline{\mathcal{P}}'_i = \sum_{i \in I} \frac{n_i \overline{\mathcal{P}}_i + k_i}{n_i + N} = \sum_{i \in I_A} \frac{n_i \overline{\mathcal{P}}_i + k_i}{n_i + N} + \sum_{i \in I_C} \frac{n_i \overline{\mathcal{P}}_i + k_i}{n_i + N}$$

We now reason on each part separately.

For $i \in I_A$ we have $\frac{k_i}{N} \leq \overline{\mathcal{P}}_i$, which also means $N \leq \frac{k_i}{\overline{\mathcal{P}}_i}$.

$$\sum_{i \in I_A} \frac{n_i \overline{\mathcal{P}}_i + k_i}{n_i + N} \quad \geq \quad \sum_{i \in I_A} \frac{n_i \overline{\mathcal{P}}_i + k_i}{\left(n_i + \frac{k_i}{\overline{\mathcal{P}}_i}\right)} = \sum_{i \in I_A} \frac{n_i \overline{\mathcal{P}}_i + k_i}{\left(\frac{n_i \overline{\mathcal{P}}_i + k_i}{\overline{\mathcal{P}}_i}\right)} = \sum_{i \in I_A} \overline{\mathcal{P}}_i \left(\frac{n_i \overline{\mathcal{P}}_i + k_i}{n_i \overline{\mathcal{P}}_i + k_i}\right) = \sum_{i \in I_A} \overline{\mathcal{P}}_i.$$

Next, for $i \in I_C$ we have $\frac{k_i}{N} > \overline{\mathcal{P}}_i$, and thus also $k_i > \overline{\mathcal{P}}_i N$. Then we have the following:

$$\sum_{i \in I_C} \frac{n_i \overline{\mathcal{P}}_i + k_i}{n_i + N} \quad > \quad \sum_{i \in I_C} \frac{n_i \overline{\mathcal{P}}_i + \overline{\mathcal{P}}_i N}{n_i + N} = \sum_{i \in I_C} \overline{\mathcal{P}}_i \frac{n_i + N}{n_i + N} = \sum_{i \in I_C} \overline{\mathcal{P}}_i.$$

Finally, we put the two partitions back together using the inequalities derived on both, and conclude by using the assumption that the prior is valid, i.e., $\sum_i \overline{\mathcal{P}}_i \geq 1$:

$$\sum_i \overline{\mathcal{P}}'_i = \sum_{i \in I_A} \frac{n_i \overline{\mathcal{P}}_i + k_i}{n_i + N} + \sum_{i \in I_C} \frac{n_i \overline{\mathcal{P}}_i + k_i}{n_i + N} \quad > \quad \sum_{i \in I_A} \overline{\mathcal{P}}_i + \sum_{i \in I_C} \overline{\mathcal{P}}_i = \sum_{i \in I} \overline{\mathcal{P}}_i \geq 1.$$

The proof for the lower bounds, that $\sum_i \underline{\mathcal{P}}'_i \leq 1$, follows the same reasoning by symmetry.

$\square$

## C   Exploration Scheme

We detail the exploration scheme in Algorithm 1 below.

## D   Detailed Problem Domain Descriptions

We give a detailed description of each model we use in the experimental evaluation below, together with the specification and (if applicable) its source.

**Chain problem.**   We consider a larger version of the chain problem Araya-López et al. [2011] with 30-states. There are three actions, one progresses with probability $0.95$ to the next state, and resets the model to the initial state with probability $0.05$. The second action does the same, but with reversed probabilities. The third action has probability $0.5$ for both progressing and resetting. Every action gets a reward of 1. As specification, we minimize the reward to reach the last state of the chain: $R_{\text{Min}}(\lozenge T)$. This can be seen as an instance of the stochastic shortest path problem [Bertsekas, 2005].

---

**Algorithm 1** Exploration scheme.

---

**Input:** $K$ : number of trajectories.
**Input:** $H$ : max trajectory length.
**Input:** $M(S, s_I, A, P, R)$ : underlying MDP.
**Input:** $\varphi$ : specification.
**Input:** $\mathcal{A}$: algorithm for exploration and robust verification.

```
 1: for s, a ∈ S × A do
 2:        ▷ Initialize counters
 3:        #(s, a) = 0
 4:        #(s, a, s') = 0 : ∀s' ∈ S
 5:        #_i(s, a) = 0
 6:        #_i(s, a, s') = 0 : ∀s' ∈ S
 7: end for
 8: for k ∈ [1, · · · , K] do
 9:     if ∃s, a ∈ S × A : #_i(s, a) >= #(s, a) then
10:          ▷ Compute new policies.
11:          Give iteration counters #_i(s, a) and #_i(s, a, s') to 𝒜
12:          Get sampling policy from 𝒜 : π_sampling
13:          Get robust policy from 𝒜 : π_robust
14:          Evaluate π_robust on M according to φ
15:          ▷ Update global counters
16:          #(s, a) += #_i(s, a) : ∀s, a ∈ S × A
17:          #(s, a, s') += #_i(s, a, s') : ∀s, a ∈ S × A × S
18:          ▷ Reset iteration counters
19:          #_i(s, a) = 0 : ∀s, a ∈ S × A
20:          #_i(s, a, s') = 0 : ∀s, a ∈ S × A × S
21:      end if
22:      Sample τ from M following π_sampling with max size H
23:      Increment #_i(s, a) and #_i(s, a, s') according to τ.
24: end for
```

---

**Aircraft collision avoidance.** We model a small, simplified instance of the aircraft collision avoidance problem Kochenderfer [2015]. Two aircraft, one controlled, one adversarial, approach each other. The controlled aircraft can increase, decrease, or stay at the current altitude with a success probability of $0.8$. The adversarial aircraft can do the same, but does so with probabilities $0.3$, $0.3$, $0.4$ respectively. To goal is to maximize the probability of the two aircraft passing each other without a collision, expressed by the temporal logic specification $\mathbb{P}_{\text{Max}}(\neg\text{collision} \cup \text{passed})$.

**Grid world.** We consider a slippery $10 \times 10$ grid world as in Derman et al. [2019], where a robot starts in the north-west corner and has to navigate towards a target position. The robot can move in each of the four cardinal directions with a success probability of $0.55$, and a probability of $0.15$ to move in each of the other directions. Throughout the grid, the robot has to avoid traps. The model has 100 states and 1450 transitions. We consider *safety* specifications that maximize the probability for reaching the north-east (NE) corner without getting trapped, i.e. $\mathbb{P}_{\text{Max}}(\neg\text{trapped} \cup \text{NE})$.

**Bandit.** We consider a 99-armed bandit Lattimore and Szepesvári [2020], where each arm (action) has an increased probability of success, $0.01, 0.02, \ldots, 0.99$ respectively. The goal is to find the action that has the highest probability of success: $\mathbb{P}_{\text{Max}}(\Diamond \text{ success})$.

**Betting game.** We consider a betting game from [Bäuerle and Ott, 2011], in which an agent starts with 10 coins and attempts to maximize the number of coins after 6 bets. When a bet is won, the amount of coins placed is doubled, when lost the amount of coins placed is removed. The agent may place bets of 0, 1, 2, 5, and 10 coins. We consider two versions of the game, one which is FAVOURABLE to the player, with a win probability of $0.8$, and one that is unFAVOURABLE to the player, with a win probability of $0.2$. After six bets the player receives a state-based reward that equals the number of coins left. This yields an MDP of 300 states and 1502 transitions. The specification is to maximize the reward after for reaching a terminal state after these six bets.

# E   Comparison with UCRL2.

UCRL2 [Jaksch et al., 2010] is a reinforcement learning algorithm, meaning it is designed to collect reward while exploring, in contrast to our setting where we have a clear separation between exploration and policy computation (i.e. computing the reward). We use UCRL2 to learn a uMDP, but then compute a robust policy on that learned model, instead of an optimistic policy as is standard for UCRL2. Furthermore, we make the following changes to the UCRL2 algorithm to make it more comparable to our setting:

1. Just as with the PAC intervals, we use MAP-estimation instead of the frequentist approach to estimate the point estimates.

2. UCRL2 does not compute individual intervals, but a set of distributions around the distribution of point estimates. In particular, it takes the set of distributions for which the $\ell^1$ norm to the estimated distribution is less than or equal to

$$\sqrt{\frac{14|S|\log(2|A|t_k \cdot {}^1\!/\gamma)}{\max(1, N)}}, \tag{5}$$

   see [Jaksch et al., 2010] for details on this bound. We use this bound for the individual intervals (bounded by $\varepsilon$ to ensure lower bounds greater than zero). As the robust value iteration for uMDPs restricts to valid distributions with the product of these intervals, we do not use the $\ell^1$ norm, but put in the intervals derived from MAP-estimation with this bound.

3. Finally, the UCRL2 algorithm assumes that the reward function is also unknown, and attempts to learn that as well. We assume the reward function to be known in our setting.

# F   Complete Experimental Evaluation

In this Section we present the full experimental evaluation of our method. For our main conclusions, we refer back to Section 6, and only comment here on additional experiments not found in the main text.

In Figure 2 we show the Performance, and Estimation Error for all six environments and all four learning methods. Additionally, we also define a *model error*: For each transition, we compute the maximum distance between the true probability and the lower and upper bounds of the interval in the uMDP (or the point estimate for MAP-estimation), and then take the average over all these distances.

Figure 1 shows the performance of LUI on the Grid environment with different prior strength choices. We note little difference, and thus conclude that our LUI approach is not very sensitive to small changes in prior choices.

Figures 3 and 4 shows the results for switching the probabilities on the Chain environment, for two different randomization parameters in the exploration: $\xi = 0.8$ and $\xi = 1.0$ (no randomization). Figures 5 and 6 show the same experiment on the Betting Game, switching from a favourable game to an unfavourable one, again with randomization parameters $\xi = 0.8$ and $\xi = 1.0$. We note the need for randomization during exploration, especially in the Chain environment.

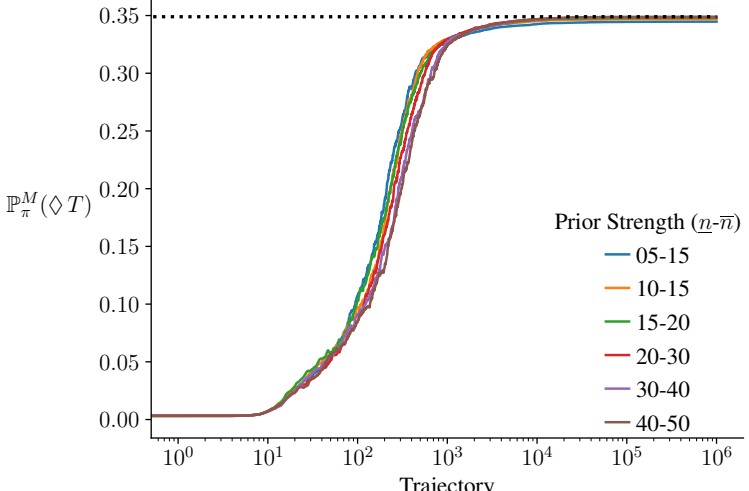

Figure 1: Prior strength evaluation on the Grid environment.

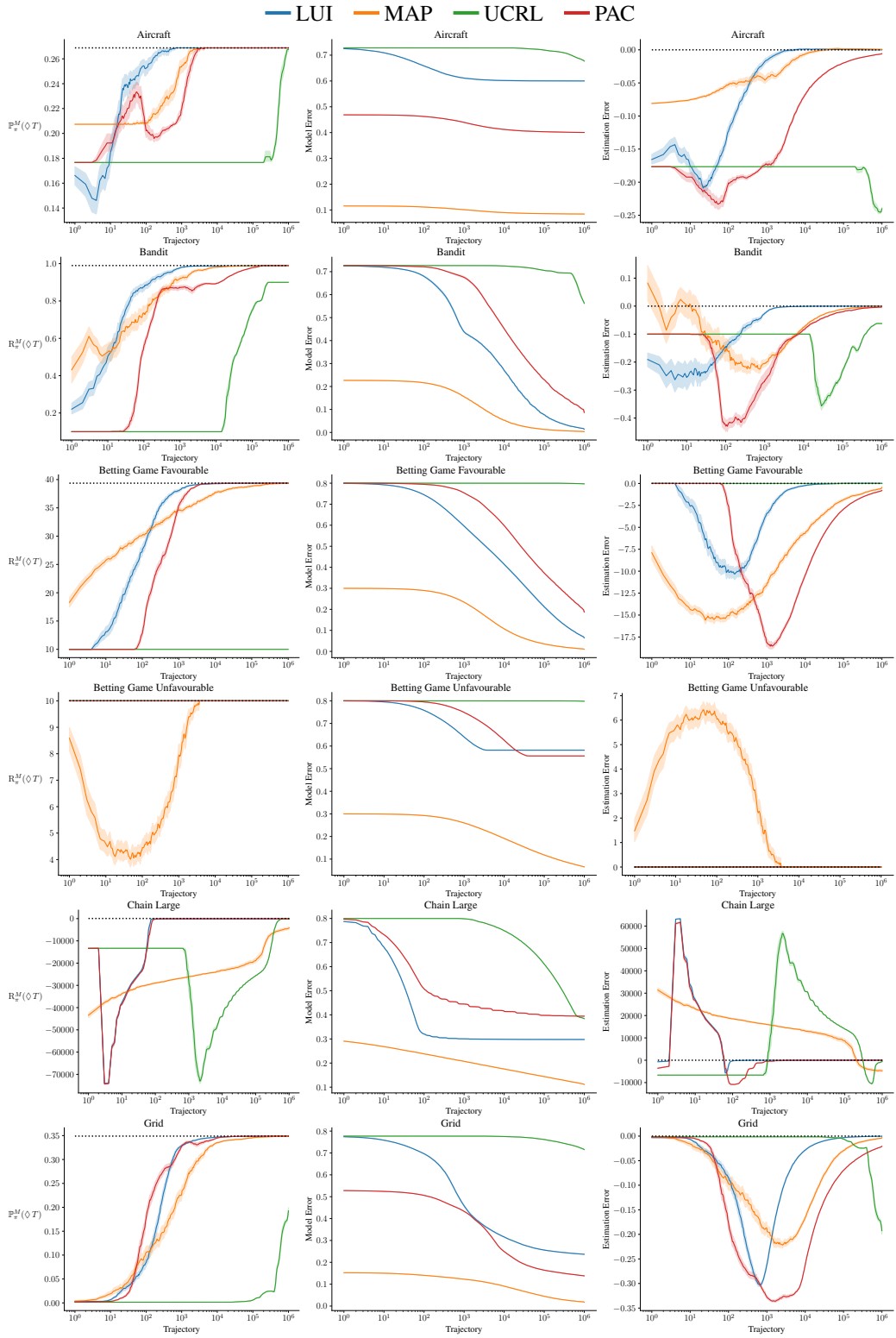

Figure 2: Performance, model error, and estimation error for all four learning methods on all six environments.

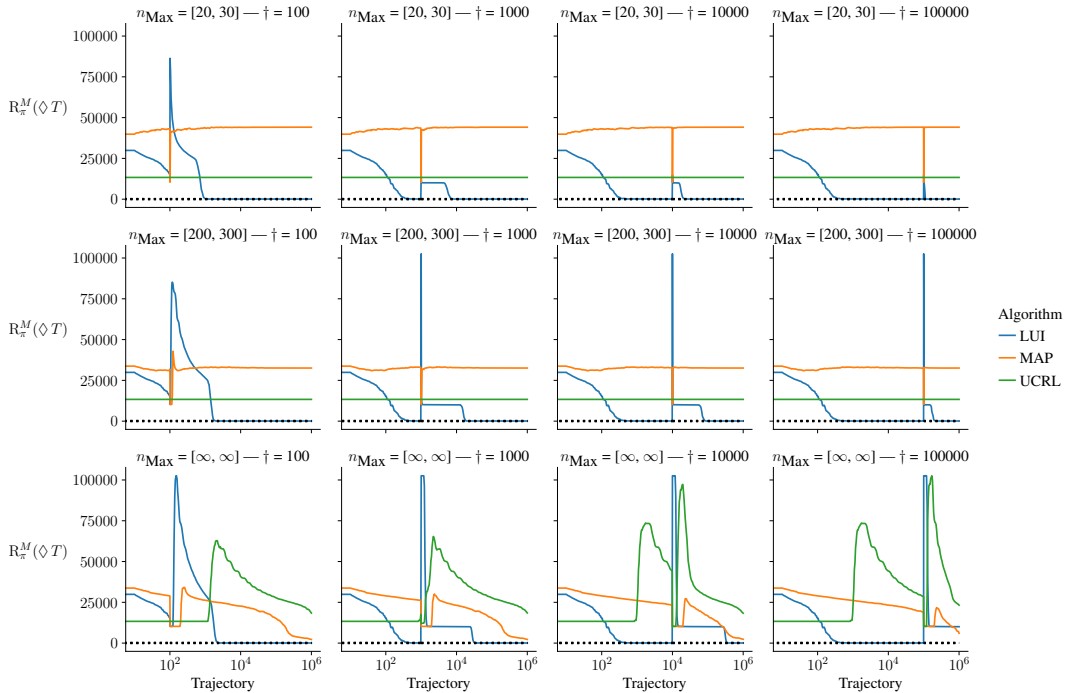

Figure 3: Environment change on the Chain environment at different points $\dagger \in \{10^2, \ldots, 10^5\}$ using randomization parameter $\xi = 0.8$ in the exploration.

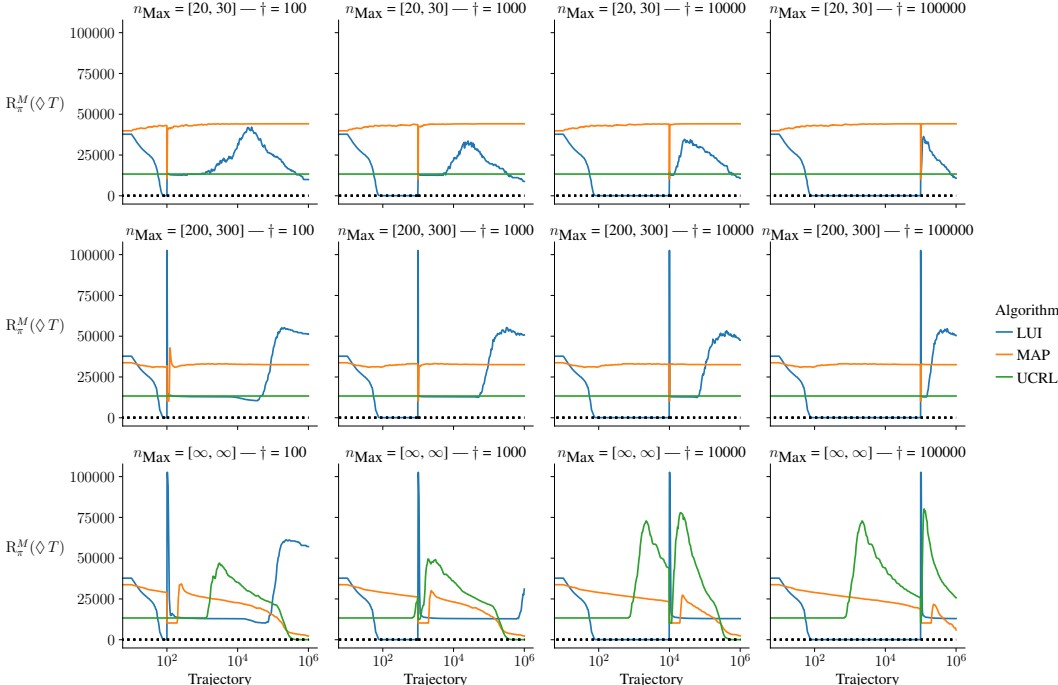

Figure 4: Environment change on the Chain environment at different points $\dagger \in \{10^2, \ldots, 10^5\}$ using randomization parameter $\xi = 1.0$ in the exploration.

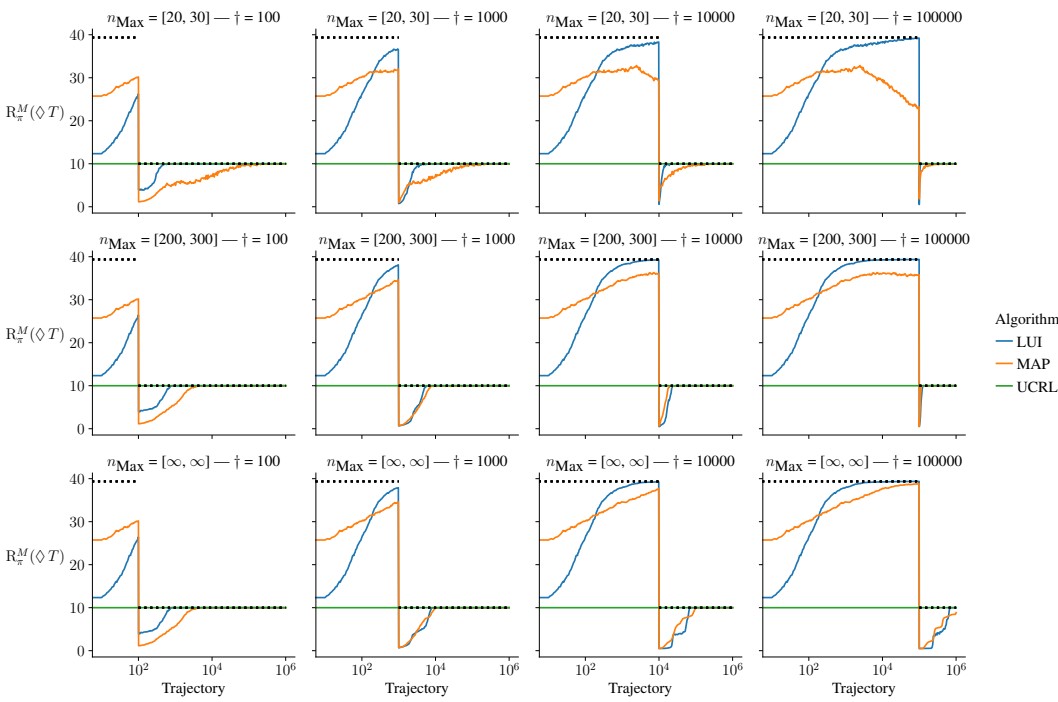

Figure 5: Environment change on the Betting Game environment at different points † ∈ $\{10^2, \ldots, 10^5\}$ using randomization parameter $\xi = 0.8$ in the exploration.

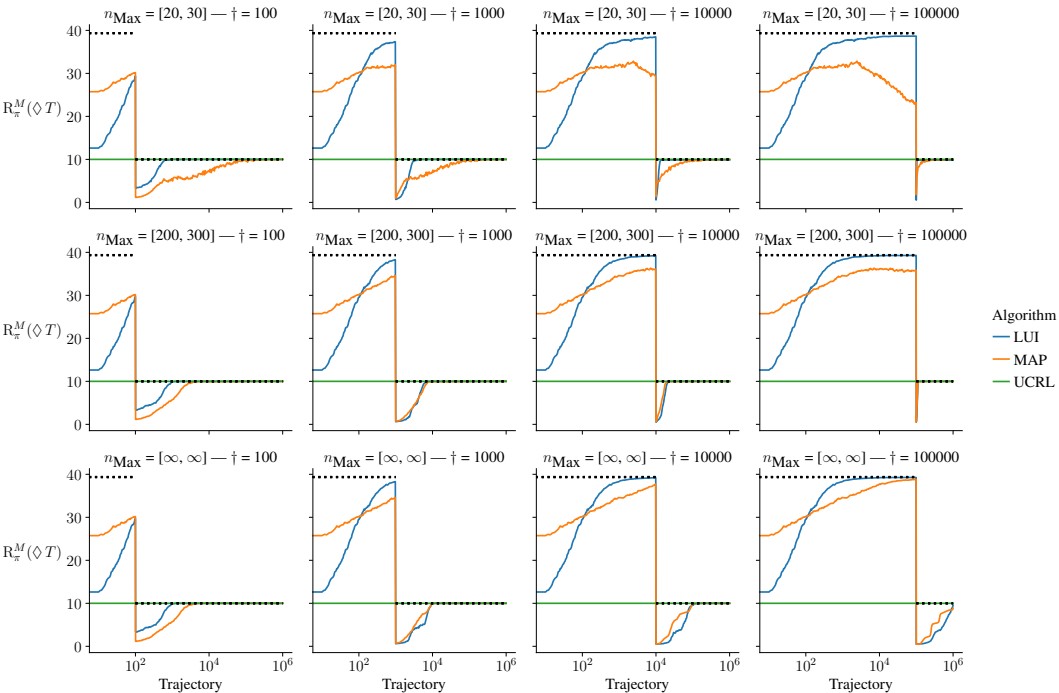

Figure 6: Environment change on the Betting Game environment at different points † ∈ $\{10^2, \ldots, 10^5\}$ using randomization parameter $\xi = 1.0$ in the exploration.