# OpenReview forum: "Robust Anytime Learning of Markov Decision Processes"
_NeurIPS.cc/2022/Conference — NeurIPS 2022 Accept_

### Official Review · Reviewer_r6yL · 2022-07-10

**Rating:** 6
**Confidence:** 4
**Soundness:** 3 good
**Presentation:** 4 excellent
**Contribution:** 2 fair

**Summary:**

The paper develops an anytime-learning technique to estimate the transition probabilities of a Markov Decision Process (MDP). Given an MDP with a known transition graph, the methodology begins by calculating the worst-case performance of a policy $\pi$ with respect to an uncertain MDP (uMDP) and an associated temporal-logic specification.  If such a policy violates the desired threshold, the authors sample a trajectory and update the uncertainty set according to either Hoeffding's inequality or linearly updating intervals. Numerical results evaluate the approach with respect to performance and estimation/model errors.

**Questions:**

1. Are there any insights/examples that illustrate when the approach leads to optimal policies? And poor policies?

2. When would LUI be preferable to PAC? While this was discussed briefly, I feel that some clarifications could be extremely helpful.

**Strengths And Weaknesses:**

Strengths
- Innovative robust perspective to learn transition probabilities
- Excellent exposition

Weaknesses
- Lack of theoretical insights
- Limited numerical analysis

Major Comments

Overall, the paper is rigorous and excellently well written; it was very enjoyable to read, with all steps clearly motivated and outlined. I found the idea to be innovative as typically probability estimations in MDPs are limited to MAP-based approaches, and there is a clear methodological contribution provided by the "robust" view that the authors offer. However, perhaps due to its "proof of concept" aspect, the paper tends to be much more on the computational side and lacks a more in-depth analysis of the approach, in my personal view. More precisely:

(i) [Theory.]  It was not clear to me why this method would ever be (rigorously) effective. Specifically, the authors mention that the learning is "conservative" due to the robust nature of the worst-case policy evaluation. While the numerical results are promising, there are few theoretical insights on how conservative it could be. Is there any particular MDP structure (e.g., in the transition probabilities or transition graph) that the policy would be optimal, i.e., the transition probabilities match the ground truth? Or, conversely, could it be arbitrarily poor, and when does that happen? What was special about the instances tested for the good performance that was observed?

Perhaps my primary concern is that there is little indication of the overall performance of this approach, and the tradeoffs associated with the robust view, which for me is one of the key (and interesting) contributions of the work.

(ii) [Numerical Analysis.] My impression is that the numerical results and the choice of environments were well carried out and sufficiently diverse. However, it would benefit from a more thorough analysis and intuition on why some of the behaviour was observed. First, why would LUI and PAC be roughly equivalent? Does that mean the method is less sensitive to the Bayesian updates that are performed? Second, results on the "Chain" environment are very interesting; I wonder why LUI/PAC are choosing the bad actions; perhaps that can relate to point (i) above? Overall, the paper is missing a more careful discussion to understand when the proposed methods work best or can be outperformed by reinforcement learning / MAP techniques.

Minor Note
- Proof *of* Proposition 1

---

> ### Author Response · Authors · 2022-08-02
> **Response to review r6yL**
>
> We thank the reviewer for their helpful questions and comments.
>
> Regarding the theory and Q1: In general, for uMDPs, we have that if all transitions $P(s,a,s’)$ of an MDP are contained in the intervals $\mathcal{P}(s,a,s’)$ of the uMDP, then the value of the robust policy for the uMDP will be less or equal to the value of the optimal policy for the MDP. Similarly, the value of the optimistic policy for the uMDP will be greater or equal to the value of the optimal policy for the MDP. Hence, the optimal value can be bounded by the values of the robust and optimistic policies. Furthermore, the smaller the intervals become, the tighter these bounds become, though determining when the intervals will be small enough such that a robust policy is already optimal, will largely depend on the model and specification.
>
> Note, however, that we cannot give an absolute guarantee that the intervals contain the true transition probabilities. Only the PAC approach offers a PAC guarantee on individual transitions, which can also be extended to a PAC guarantee on the value of the (robust) policy, see:
>
> Ashok, P., Křetínský, J., & Weininger, M. (2019, July). PAC statistical model checking for Markov decision processes and stochastic games. In International Conference on Computer Aided Verification (pp. 497-519). Springer, Cham.
>
> Regardless of the method used (LUI or PAC), the intervals will converge to point intervals of the exact true probability of the underlying MDP in the infinite run. As a consequence, the robust policy will eventually be optimal.
>
> Regarding the numerical analysis and Q2: The most likely reason for LUI and PAC giving similar results is that both methods learn similar intervals. In particular, it should be noted that different intervals may still lead to the same robust policy and thus the same value. On most models, LUI has a higher model error compared to PAC, meaning most of the intervals are larger than PAC, yet achieves similar performance. This implies that LUI can better concentrate the learning effort on transitions that matter the most for optimal performance. Finally, LUI also has an estimation error that is smaller than that of PAC in some models, and equivalent in others. This means that LUI is better at predicting its own performance based on the learned model than PAC.
>
> As for the Chain environment specifically, in this environment, it is especially hard to get a decent number of samples of transitions later on in the chain due to our ‘online’ setting where we may only sample trajectories and not individual actions. As such, it is easy to estimate a couple of transitions wrong, leading to choosing the incorrect action at those states. As the reward function counts the number of steps to the last state in the chain and choosing the wrong action has a high probability of resetting the agent to the initial state, the reward quickly accumulates as the agent has to try the bad action an enormous number of times before it gets lucky and moves on to the next state. Note, however, that all methods suffer from the same problem, and that LUI and PAC are the first to find the optimal policy within a reasonable number of processed trajectories (~100 versus $10^5$ for MAP and UCRL2). We have updated the paper to reflect these observations a bit better. See lines 336-339 and 344-350.

---

> > ### Comment · Reviewer_r6yL · 2022-08-09
> > **Feedback**
> >
> > Thank you for the clarification, those were really insightful points and I appreciate the thorough response.

---

### Official Review · Reviewer_CuSo · 2022-07-11

**Rating:** 6
**Confidence:** 3
**Soundness:** 3 good
**Presentation:** 3 good
**Contribution:** 2 fair

**Summary:**

This paper considers the problem of learning an (over) approximation Markov Decision Process (MDP) in the form of a uncertain Markov Decision Processes (uMDP). This model is then used synthesizing a policy that is robust to the inferred uncertainty in the transition probabilities. The is done as an iterative process: (i) A uMDP is conjectured. (ii) An (optimistic) exploration policy is synthesized (iii) The transition probability intervals are updated based on the newly collected data. This update is either done using a PAC estimate on the interval or a form of weighted updating of the bounds (called linear updating). The paper ends by comparing against a MAP base-line and an established robust RL algorithm.

**Questions:**

1. How is the PAC result employed given the distribution shift? Is this update procedure not iterative?
1. Can the minimal transition probability $\epsilon$ be made symbolic, e.g. in infinitesimal ? In particular, it feels like one could easily detect if $\epsilon$ should have been made smaller since its only used to clip the probability distributions right?
1. Aside from capturing model uncertainty, another application of interval MDPs is as a form of model compression. Is there a path for making the approach work in such settings, i.e., trade-off resolution for a smaller (easier to train) model?

**Limitations:**

- Limited to finite MDPs where the graph is already known. This is often satisfied, but there are interesting domains, e.g., continuous, where this does not hold.
- Not as fast as robust RL baseline at adapting to distribution shift. This is understandable and listed as future work.

**Strengths And Weaknesses:**

# Strengths
1. The problem is well motivated and of interest to this venue.
1. The approach builds off of well established theoretical machinery and combines them in a well motivated way.
1. The baselines and metrics seem reasonable and help illustrate the strength of the approach.

# Weaknesses
1. (minor) In the formal verification community, this model is also sometimes know Interval Markov Decision Process. It would have been helpful early on to mention this.
1. Need to know a-priori which transitions are impossible. This seems reasonable in many domains, but limits the applicability.
1. Unclear how this approach scales to very large state and action spaces.
1. Arguably an incremental result that builds on well established machinery.
1. (minor) Does not easily detect that there has been a distribution shift (change of environment). This is left as future work.
1. **(primary concern)** I did not understand how the PAC bound was used iteratively. Doesn't Hoeffding's result require i.i.d. sampling? This is violated by changing the exploration policy right? This is my primary hesitation in assigning a higher score.

---

> ### Author Response · Authors · 2022-08-02
> **Response to review CuSo**
>
> We would like to thank the reviewer for their comments and questions.
>
> Indeed, uncertain MDPs are known under several different names, like interval MDPs or robust MDPs, in other communities. We have clarified this point in the introduction, line 31.
>
> Regarding i.i.d. sampling: The PAC guarantee is for individual transitions $(s, a, s’)$. Since the transition function is Markovian, and we observe both states and actions, it does not matter how we gain access to the state-action pair $(s, a)$, thus changing the exploration policy does not violate i.i.d. sampling of $(s, a)$. See also the first paragraph of Section 5 in the paper.
>
> Q1: In the case of a distribution shift, the PAC result loses its formal guarantee and should only be considered as a way to construct intervals without a formal guarantee on their correctness. We added a remark in lines 369-371.
>
> Q2: Yes, the $\epsilon$ is only used to keep the lower bound of the intervals away from zero, and could be made symbolic. We clarified this in line 196.
>
> Q3: While we are unaware of any specific paper using uncertainty as a form of model compression, it makes sense that one can remove a feature from the state-space in exchange for uncertainty on the transitions. In the context of learning, however, the underlying model one tries to learn would then be a uMDP. Our method, in its current form, is not suited for learning uMDPs, as the intervals converge to the ground truth probabilities. To converge to proper intervals instead, one would need a decent stopping criterion, like a minimal size for each interval.

---

> > ### Comment · Reviewer_CuSo · 2022-08-04
> > **Thank you for the clarifications**
> >
> > I indeed missed the fact that the distribution indexed by state and action is independently drawn due to the Markov assumption.
> >
> > Thank you for the clarifications.

---

### Official Review · Reviewer_2Job · 2022-07-12

**Rating:** 4
**Confidence:** 3
**Soundness:** 3 good
**Presentation:** 3 good
**Contribution:** 2 fair

**Summary:**

The paper proposes a method for learning the probabilities of the transitions a Markov Decision Process from observed data. The method assumes that the structure of the MDP is available to it. It initializes these probabilities to a wide interval, and then as data keeps arriving, the probability is updated. Based on the learned probabilities, the authors compute an optimal policy, and learning stops once the optimal policy rewards reach a certain threshold (or may be stopped by the user after a certain amount of data is gathered).

**Questions:**

Page 3: The specification of P_MaxMin assumes that the MDP is chosen from the possible MDPs and the best policy is chosen for it. How is the MDP chosen --- is it assumed that the MDP is chosen so that the best policy for it will produce the smallest reward?

Page 6: It was not clear to me how to initialize prior strength interval n_i. You said that they can be chosen arbitrarily, so one choice could be [0,0]. But this would violate Proposition 2 --- if P_i = (e, 1-e), then if N = 1, k_i = 0, then P'_i = (0,0). There was no proof in Appendix B.

**Limitations:**

Not applicable.

**Strengths And Weaknesses:**

The paper is clearly written for the most part. The computation of the P_MinMax was not shown in the paper, but I am assuming this was already described in the papers cited.

The algorithm is a rather straightforward generalization of estimating a probability from sample data --- given a stream of 0, 1 values, what is a good interval bound for the probability? The authors propose either using Hoeffding's inequality, or a variant of Laplace smoothing. It would be good to understand how the MDP structure helps in getting these estimates quickly, as of now it seemed that the different transitions were treated independently.

---

> ### Author Response · Authors · 2022-08-02
> **Response to review 2Job**
>
> We would like to thank the reviewer for their comments and questions.
>
> Indeed, computation of policies and values satisfying robust or optimistic specifications, like $P_{MinMax}$ is well established; see e.g.
>
> Wolff, E. M., Topcu, U., & Murray, R. M. (2012, December). Robust control of uncertain Markov decision processes with temporal logic specifications. In 2012 IEEE 51st IEEE Conference on Decision and Control (CDC) (pp. 3372-3379). IEEE.
>
> and
>
> Puggelli, A., Li, W., Sangiovanni-Vincentelli, A. L., & Seshia, S. A. (2013, July). Polynomial-time verification of PCTL properties of MDPs with convex uncertainties. In International Conference on Computer Aided Verification (pp. 527-542). Springer, Berlin, Heidelberg.
>
> as also cited in the paper.
>
> Regarding MDP structure: If further structural knowledge is assumed, for example, if it is known that certain transitions have the same probability p, one can easily exploit this knowledge by merging the data for these transitions to get a single estimate of p. More complex dependencies may also be exploited, like arbitrary linear dependencies. This would require solving a linear equation system, which, if the number of dependencies grows large, may slow down the learning.
>
> Q page 3: Yes, the MDP is chosen adversarially. A uMDP can be seen as a two-player stochastic game where one player (the agent) tries to maximize the reward (or more generally, specification) by choosing the actions, and an adversary (or the world) chooses the MDP in an attempt to minimize the reward (or specification). Such MaxMin (or MinMax) problems are well established for uMDPs. See the references above and in the paper.
>
> Q page 6: This is a slight error in the text, thanks to the reviewer for spotting it. The strengths $n_i$ can be chosen arbitrarily $\geq 1$. Furthermore, the $n_i$ could be seen intuitively as pseudo-counts, that is, a (fictitious) number of samples used to base the prior interval on. Higher values for $n_i$ put more strength on the prior, expressing that we believe the chosen prior is ‘good’, while lower values express less confidence in the prior. Since we suggest taking prior intervals $[\epsilon, 1-\epsilon]$, lower values for the strengths are more reasonable in this case as the interval captures almost all possible probabilities. We have corrected this error in lines 250-251.

---

### Author Response · Authors · 2022-08-02
**Rebuttal version of the paper**

We would like to thank the reviewers for their helpful questions and comments.
We have incorporated the most important points from the reviews into the rebuttal version of the paper, and point to the lines that have been changed in the response to each individual review below.

---

### Meta-Review · Area_Chair_ZDrg · 2022-08-24

**Recommendation:** Accept
**Confidence:** Less certain

**Metareview:**

The paper makes a good algorithmic contribution and provides its theoretical analysis. It would benefit from a more extensive empirical evaluation, for instance, evaluating on IPPC-style MDPs expressed in PDDL and RDDL and on MDPs with high-dimensional observations (block MDPs), but even as is this work is a solid step forward.

**Award:**

No

---

### Decision · Program_Chairs · 2022-09-14

Accept